# Deletions linked to *PROG1* gene participate in plant architecture domestication in Asian and African rice

Yongzhen Wu[1,3], Shuangshuang Zhao[1], Xianran Li[1,4], Bosen Zhang[1,5], Liyun Jiang[1], Yanyan Tang[1], Jie Zhao[1], Xin Ma[1], Hongwei Cai [1], Chuanqing Sun[1,2] & Lubin Tan [1]

Improving the yield by modifying plant architecture was a key step during crop domestication. Here, we show that a 110-kb deletion on the short arm of chromosome 7 in Asian cultivated rice (*Oryza sativa*), which is closely linked to the previously identified *PROSTRATE GROWTH 1* (*PROG1*) gene, harbors a tandem repeat of seven zinc-finger genes. Three of these genes regulate the plant architecture, suggesting that the deletion also promoted the critical transition from the prostrate growth and low yield of wild rice (*O. rufipogon*) to the erect growth and high yield of Asian cultivated rice. We refer to this locus as *RICE PLANT ARCHITECTURE DOMESTICATION* (*RPAD*). Further, a similar but independent 113-kb deletion is detected at the *RPAD* locus in African cultivated rice. These results indicate that the deletions, eliminating a tandem repeat of zinc-finger genes, may have been involved in the parallel domestication of plant architecture in Asian and African rice.

[1] National Center for Evaluation of Agricultural Wild Plants (Rice), MOE Laboratory of Crop Heterosis and Utilization, Department of Plant Genetics and Breeding, China Agricultural University, Beijing 100193, China. [2] State Key Laboratory of Plant Physiology and Biochemistry, China Agricultural University, Beijing 100193, China. [3] Present address: College of Agriculture, Ludong University, Yantai 264025, China. [4] Present address: Department of Agronomy, Iowa State University, Ames, IA 50011, USA. [5] Present address: Department of Crop Sciences, University of Illinois, Urbana, IL 61801, USA. These authours contributed equally: Yongzhen Wu, Shuangshuang Zhao, Xianran Li.  Correspondence and requests for materials should be addressed to L.T. (email: tlb9@cau.edu.cn)

Humans began to domesticate wild plants on different continents almost ten thousand years ago to provide food and materials, thereby promoting the development of human society[1–3]. Several genetic factors controlling key domestication-related traits, such as seed shattering[4–6], pericarp color[7], panicle architecture[8,9], and awn[10–14], have been characterized in rice (*Oryza sativa* L.). These studies provide important insights into molecular mechanisms and evolutionary trajectories underlying rice domestication. Plant architecture was one of the initial selection targets in crop domestication because of the potential advantages for increasing planting density, photosynthesis efficiency and apical dominance, and increasing grain yield. Two primary genetic contributors to plant architecture during domestication have been characterized, *teosinte branch 1* (*tb1*) gene regulating the axillary branch development in maize (*Zea mays* ssp. *mays*)[15,16] and *PROSTRATE GROWTH 1* (*PROG1*) gene controlling tiller angle and tiller number in rice[17,18]. The favorable *tb1* and *prog1* were strongly selected during the domestication of maize[19] and rice[20,21], respectively. Hence, further identification of novel genetic factors associated with plant architecture domestication will gain new insights into the history of crop domestication.

In this study, we identify a 110-kb deletion on chromosome 7, which also participates in the plant architecture domestication in Asian cultivated rice. The deletion next to *PROG1* gene harbors a tandem repeat of zinc-finger genes controlling plant architecture in wild rice. Therefore, this locus is labeled as the *RICE PLANT ARCHITECTURE DOMESTICATION* (*RPAD*) locus. Moreover, a similar but independent 113-kb deletion is detected at the *RPAD* locus in African cultivated rice, indicating that a common mechanism might be shared in the parallel domestication of plant architecture in both Asian and African cultivated rice.

## Results

### Fine mapping of a QTL for plant architecture in *O. rufipogon*.
Previous report revealed that the identical mutations of *prostrate growth 1* (*prog1*) were selected to achieve better plant architecture in Asian cultivated rice[18]. Meanwhile, 14 of 30 accessions of wild rice with prostrate growth carried identical or similar alleles as *prog1* in cultivated rice, which implied that other genes contribute to the transition of plant architecture[18]. To isolate genes associated with this transition in plant architecture during rice domestication, we identified an introgression line (DIL29) that displayed the semi-prostrate phenotype. DIL29 carries four genomic segments introgressed from *O. rufipogon* (accession DXCWR) on chromosomes 1, 4, 7, and 8 in a genetic background of *O. sativa* ssp. *indica* variety Guichao 2 (GC2) (Fig. 1a–d). Compared with the recipient parent GC2, DIL29 had a significantly greater tiller angle and tiller number, and a significantly lesser plant height, number of primary branches, number of secondary branches, and grain number, and accordingly, lower grain yield (Fig. 1e–k and Supplementary Fig. 1) ($P < 0.01$, two-tailed $t$ test). These results indicate that the *O. rufipogon* introgression segments in DIL29 might harbor a gene regulating rice plant architecture and grain yield.

To identify the quantitative trait locus (QTL) responsible for these phenotypic differences, we constructed an $F_2$ population with 317 individuals derived from a cross between DIL29 and GC2 and mapped a QTL for tiller angle, which we refer to as *SEMI-PROSTRATE GROWTH 1* (*SPROG1*), between molecular markers RM427 and M83 on the short arm of chromosome 7 (Supplementary Fig. 2). In addition to tiller angle, *SPROG1* was also associated with plant height, panicle number per plant, grain number per panicle, and grain yield per plant (Supplementary Fig. 2). We then identified informative plants with recombination

occurred in the *SPROG1* region from 3382 $F_2$ plants. Evaluation of the phenotypes of homozygous recombinants delimited *SPROG1* into an 8.9-kb interval between markers F43 and ID52 in GC2 genome (Fig. 2a). Further comparison revealed that *PROG1* is located upstream of marker F43 and therefore is not involved in the *SPROG1* fine-mapped region (Fig. 2b). This suggests that *SPROG1* is a locus involved in the transition of plant architecture during Asian rice domestication and is genetically linked with *PROG1*.

### A deletion involves in plant architecture transition in rice.
To investigate polymorphisms between the two parents (DIL29 and GC2), we screened a genomic bacterial artificial chromosome (BAC) library of DXCWR, and identified two clones (DX03A19 and DX33D19) that spanned the fine-mapped region. Sequencing and assembling two BAC clones revealed that a 110-kb chromosomal segment is deleted in GC2 relative to DXCWR (Fig. 2b). Integrating bioinformatics annotation and RNA-seq transcriptomes showed that the fine-mapped interval within the DXCWR genome (118.7 kb) comprised seven putative single $Cys_2$-$His_2$ ($C_2H_2$) zinc-finger protein-coding genes (*ZnF*), four hypothetical genes, seven transposable elements, and a remaining large proportion (54.7%) of interspersed repeat sequences (Fig. 2b, Supplementary Tables 1 and Supplementary Data 1). These putative zinc-finger proteins showed high similarity with PROG1 and contained two conserved domains, including the plant-specific QALGGH motif[22] and an ethylene-responsive element binding factor-associated amphiphilic repression (EAR) motif at the C-terminal region[23] (Supplementary Fig. 3). These results suggest that the chromosomal deletion, eliminating a tandem repeat of zinc-finger genes, might also involve in the plant architecture transition from wild to cultivated rice during domestication.

Due to the 110-kb deletion, recombinant gametes could not be generated in the *SPROG1* fine-mapped region to specify the underlying gene. This was instead done by generating 11 complementary constructs (CP-ZnF2−CP-ZnF8 and CP-OFG1−CP-OFG4) targeting each protein-coding gene within the fine-mapped region of *SPROG1*, including seven zinc-finger genes and four hypothetical genes (Fig. 2c and Supplementary Table 2). Eleven complementary constructs and one empty vector were introduced into the *O. sativa* ssp. *japonica* variety Zhonghua 17 (ZH17). The transgenic lines carrying *ZnF5*, *ZnF7*, or *ZnF8* genes had a significant increase in both tiller angle and tiller number and a dramatic decrease in both grain number per panicle and grain yield per plant (Fig. 2d–h) ($P < 0.01$, two-tailed $t$ test). By contrast, the transgenic lines produced with the empty vector and the other constructs all had an erect growth phenotype (Fig. 2d–h and Supplementary Fig. 4) ($P < 0.01$, two-tailed $t$ test). Therefore, our findings demonstrated that, among the genes eliminated by the deletion, *ZnF5*, *ZnF7*, and *ZnF8* are functional zinc-finger genes regulating plant architecture in wild rice.

### Molecular functions of three zinc-finger genes.
To investigate whether ZnF5, ZnF7, and ZnF8 were involved in transcription regulation, we conducted subcellular localization and transcriptional activation assays. We found that the ZnF5-, ZnF7-, and ZnF8-GFP fusion proteins were specifically localized to the nucleus in rice protoplasts (Fig. 3a), and that ZnF5, ZnF7, and ZnF8 exhibited strong transcriptional repression through the EAR motif (Fig. 3b). To study the temporal and spatial expression patterns of *ZnF5*, *ZnF7*, and *ZnF8*, we further analyzed their expression levels in ten tissues of *O. rufipogon* DXCWR. The transcripts of *ZnF5*, *ZnF7*, and *ZnF8* were consistently detected in tiller bases (Fig. 3c). Furthermore, the mRNA in situ

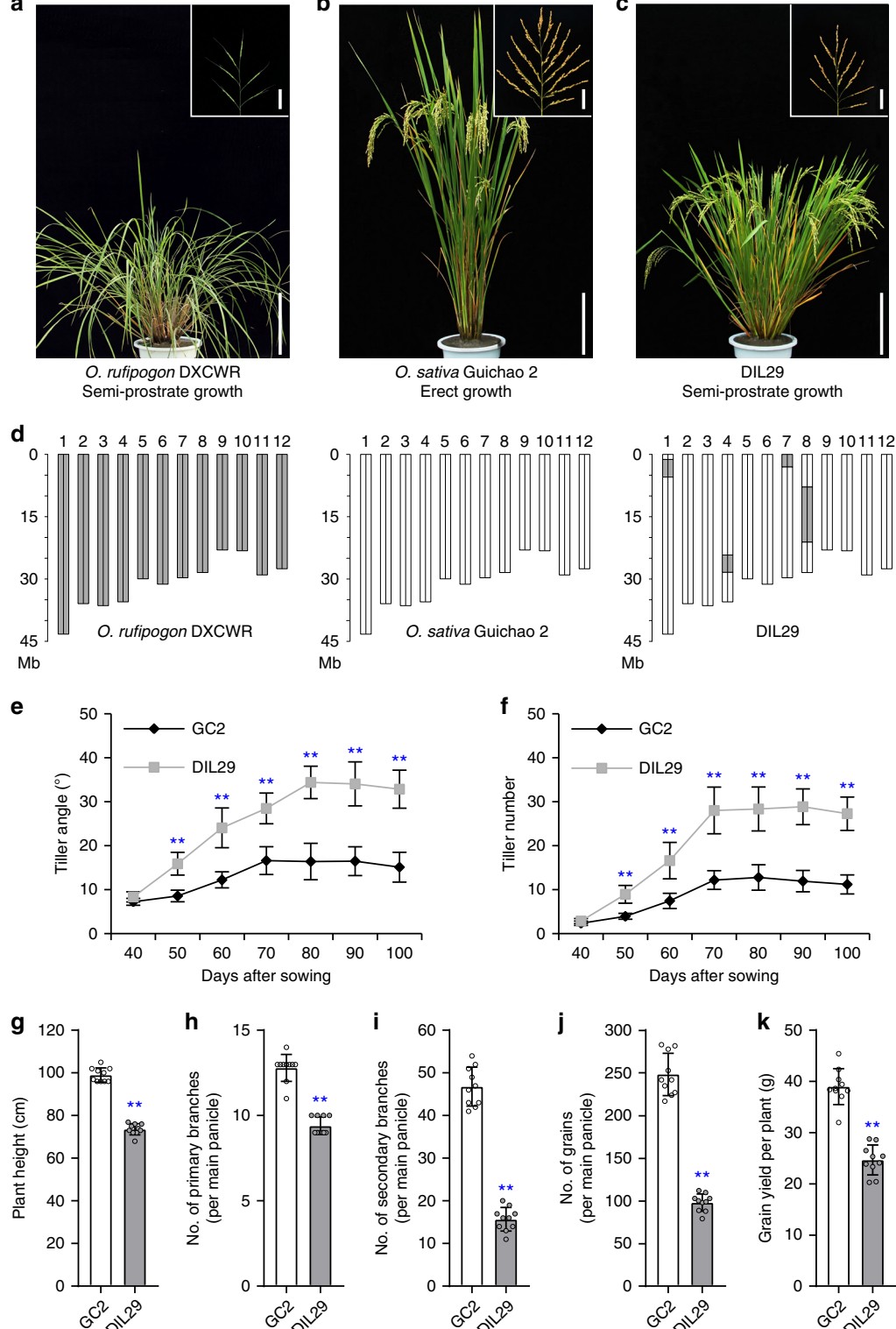

**Fig. 1** Phenotypes and genotypes. **a–c** Phenotypes of *O. rufipogon* accession DXCWR, *indica* variety Guichao 2 (GC2), and the introgression line DIL29, respectively. Panicles of DXCWR, GC2, and DIL29 are illustrated in the respective upper right corners. Scale bars, 20 cm (plant) and 5 cm (panicle). **d** Graphical genotypes. The gray regions indicate the regions that are homozygous for the DXCWR genome. The white regions indicate the regions that are homozygous for the GC2 genome. **e**, **f** Comparison of tiller angle and tiller number of GC2 and DIL29 at different developmental stages. **g–k** Comparison of plant height, number of primary branches, number of secondary branches and grain number on the main panicle, and grain yield per plant in GC2 and DIL29. Data are means ($n = 10$), with error bars showing standard deviation. Two-tailed Student's *t* tests were performed between GC2 and DIL29 (**P < 0.01)

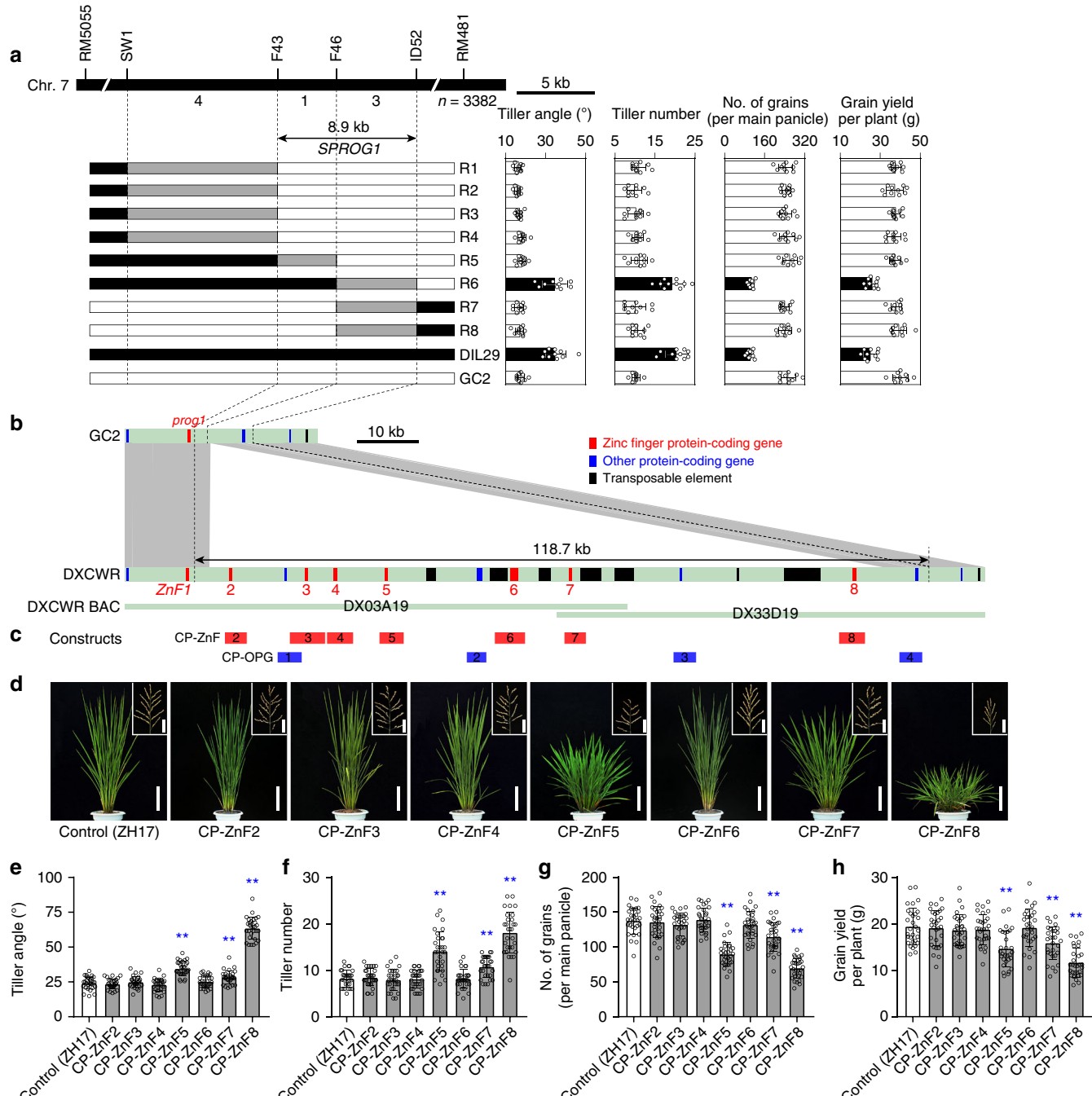

**Fig. 2** Map-based cloning of *SPROG1*. **a** *SPROG1* was delimited to an 8.9 kb region between the F43 and ID52 markers by evaluating the phenotypes of homozygous recombinants (R1 through R8). Numbers below the bar indicate the number of recombinants between the two adjacent markers. In graphical genotypes of recombinants, the black, white, and gray regions represent homozygous for the DXCWR genome, homozygous for GC2 genome, and the interval in chromosome where crossover took place, respectively. Data are means ($n = 10$), with error bars showing standard deviation. **b** Comparison of genomic sequence and gene annotation between GC2 and DXCWR in the fine-mapping region of *SPROG1*. The red, blue, and black boxes respectively indicate zinc-finger protein-coding genes, other protein-coding genes, and transposable elements. The gray shading represents the regions sharing sequence collinearity between GC2 and DXCWR genomic sequences. **c** The location of the sequences included in the 11 complementary constructs (CP-ZnF2 through CP-ZnF8 and CP-OPG1 through CP-OPG4). **d** Phenotypes of the transgenic plants of seven zinc-finger genes (*ZnF2–ZnF8*) and the control plant (ZH17). Scale bars, 20 cm (plant) and 5 cm (panicle). **e–h** Comparison of the tiller angle, tiller number, number of grains on the main panicle, and grain yield per plant between transgenic plants of seven zinc-finger genes (*ZnF2–ZnF8*) and the control plant (ZH17). Data are means ($n = 30$), with error bars showing standard deviation. Two-tailed Student's *t* tests were performed between ZH17 and transgenic plants ($^{**}P < 0.01$)

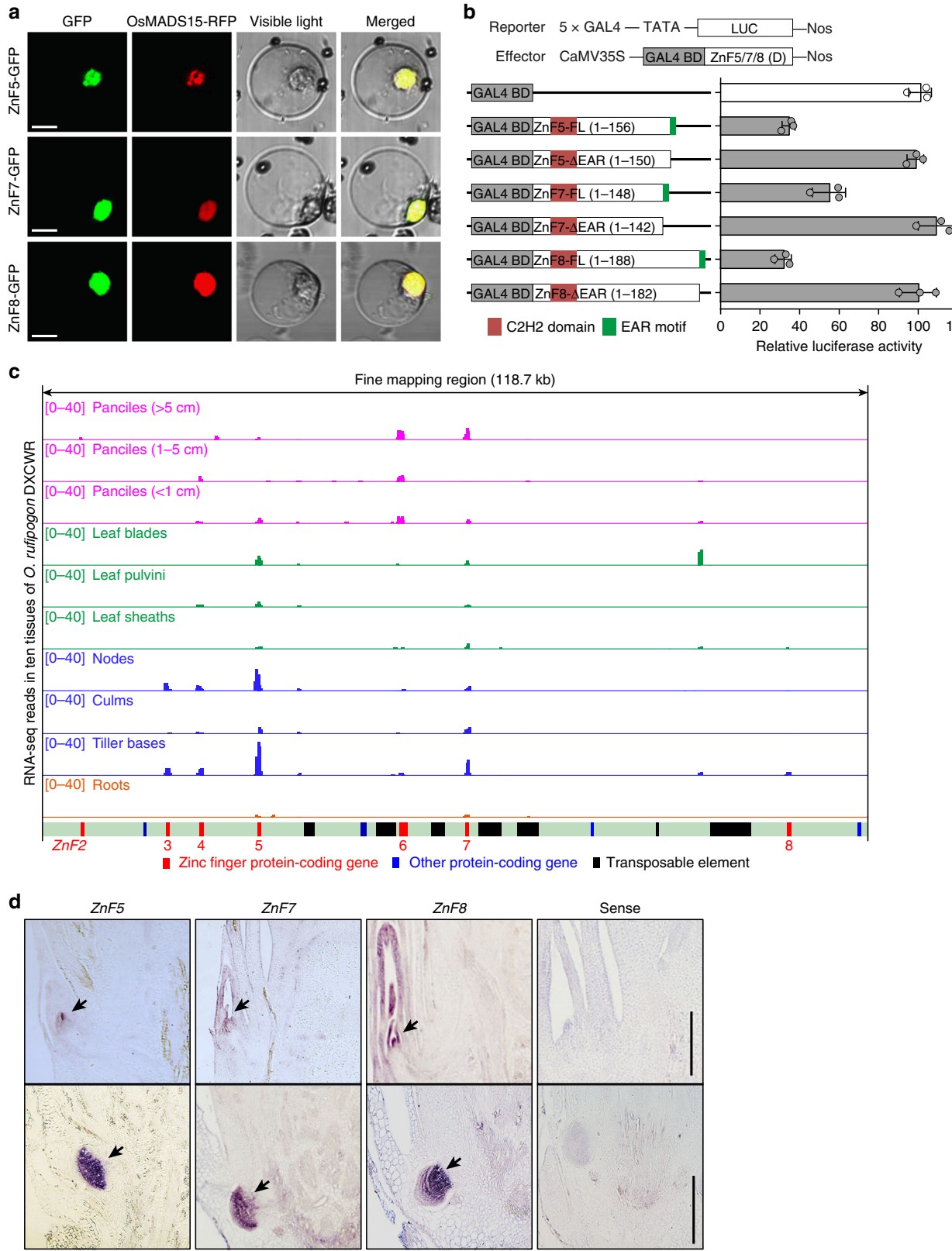

**a** GFP, OsMADS15-RFP, Visible light, Merged — ZnF5-GFP, ZnF7-GFP, ZnF8-GFP

**b** Reporter: 5 × GAL4 — TATA — LUC — Nos; Effector: CaMV35S — GAL4 BD — ZnF5/7/8 (D) — Nos

C2H2 domain; EAR motif; Relative luciferase activity

**c** Fine mapping region (118.7 kb); RNA-seq reads in ten tissues of *O. rufipogon* DXCWR
[0–40] Pancicles (>5 cm); [0–40] Pancicles (1–5 cm); [0–40] Pancicles (<1 cm); [0–40] Leaf blades; [0–40] Leaf pulvini; [0–40] Leaf sheaths; [0–40] Nodes; [0–40] Culms; [0–40] Tiller bases; [0–40] Roots

*ZnF2* 3 4 5 6 7 8

Zinc finger protein-coding gene; Other protein-coding gene; Transposable element

**d** *ZnF5* *ZnF7* *ZnF8* Sense

hybridization revealed that the transcripts of *ZnF5*, *ZnF7*, and *ZnF8* were detected in axillary bud and root primordial (Fig. 3d), which is consistent with their presumed roles in controlling tiller development and plant architecture in rice.

To analyze the molecular functions of these *ZnF* genes, we surveyed the transcriptomes of the transgenic plants of three functional *ZnF* genes (*ZnF5*, *ZnF7*, and *ZnF8*) and *PROG1* (*YJ-ZnF1*). We identified a total of 4431 genes that had different expression level (fold change ≥ 2; $P < 0.001$, negative binomial test) between at least one *ZnF* transgenic plant and the controls (ZH17) (Supplementary Data 2). Cluster analysis of the 4431 differentially expressed genes (DEGs) showed that the similarity

**Fig. 3** Transcriptional activity and expression pattern of three zinc-finger genes. **a** Subcellular localization of ZnF5-, ZnF7-, and ZnF8-GFP fusion proteins in rice protoplasts. A nuclear marker protein, OsMADS15, fused with RFP, was used as the positive control. Scale bars, 10 μm. **b** Transcriptional activity assay. The GAL4-BD fusion effectors were constructed using the entire coding region or 3′ truncated (EAR motif deletion) coding region of ZnF5, ZnF7, and ZnF8, respectively. Renilla luciferase reported gene was used as the internal control. Horizontal gray bars show the normalized mean ($n = 3$ replicates) for each construct, with error bars showing standard deviation. **c** Expression profiles of the putative genes in the fine-mapped region of SPROG1, as assessed using RNA-seq. **d** RNA in situ hybridization. Expression patterns of ZnF5, ZnF7, and ZnF8 were measured in the tiller bases at 30 days after sowing. The sense probe was hybridized and used as the negative control. Black arrowheads indicate the position of axillary bud and root primordial in the tiller base. Scale bars, 100 μm

---

is particularly high between ZnF5 and ZnF7 transgenic plants (Supplementary Fig. 5a), which is in accordance with phylogenetic analysis of these ZnF protein sequences (Supplementary Fig. 3b). In addition, a substantial amount of DEGs (approximately 69.3%) were uniquely regulated by individual ZnF genes, and 85 DGEs were commonly regulated by all four functional ZnF genes (Supplementary Fig. 5b, c), suggesting that the functional ZnF genes might regulate plant architecture through a shared pathway, but also have specific functions in regulating other pathways. Further gene ontology (GO) analysis of 584 up-regulated and 449 down-regulated DEGs by both ZnF5 and ZnF7 showed that these DEGs were enriched in multiple biological processes, including transcription regulator activity, biosynthesis of secondary metabolites, stimulus response, carotenoid biosynthesis, and plant hormone signal transduction (Supplementary Fig. 5d). Overall, these functional ZnF genes, as transcription factor, play important roles in diverse regulatory processes in rice plant architecture development.

**The deletion is a single event in Asian rice domestication.** Previous studies detected a selective sweep in 2.4−3.0 Mb on chromosome 7 (the rice reference genome IRGSP 4.0) [21], which spans both the domestication-related gene PROG1 and the deletion site. To elucidate the underlying associations between the chromosomal deletion and plant architecture domestication in rice, we sequenced the flanking genomic region covering the deletion site in 133 Asian rice cultivars (56 indica and 77 japonica cultivars collected from 16 countries). We found that all surveyed cultivars had an identical sequence at the deletion site (Fig. 4a). We further aligned the short reads from 1082 accessions of cultivated rice[21] to the chromosomal region surrounding the deletion site. We found that 717 accessions had the properly paired-end reads that were uniquely positioned at this region, supporting the deletion event. In addition, the consensus sequence of all reads covering the deletion site from 412 accessions, including 192 indica, 8 aus, 174 temperate japonica, 16 tropical japonica, 4 aromatic, and 18 intermedia ecotype cultivars, is the same as what we identified in 133 cultivars (Supplementary Fig. 6). Altogether, these results suggested that the deletion might be a single evolutionary event that occurred during the domestication of Asian cultivated rice.

We examined the nucleotide diversity of 14 sites in an approximately 1-Mb genomic region surrounding the deletion site and detected a 500-kb selective sweep with a dramatic reduction of relative nucleotide diversity in cultivated rice (Fig. 4b), which was consistent with the previous report[21]. Notably, the fixation index ($F_{ST}$) between japonica and indica cultivars was significantly lower in the selective sweep than that detected in its flanking regions (Fig. 4c), implying that this region, covering PROG1 gene and the deletion, had undergone directional selection during Asian cultivated rice domestication.

To trace the evolutionary trajectory of the deletion event, we conducted phylogenetic analysis using the DNA sequences from three sites: Sites 2 and 13 were the flanks of the selective sweep, and Site 7 was at the center of the region, as shown in Fig. 4. The results showed that at Site 7, all cultivated rice formed a single clade with only minor sequence differences (Supplementary Fig. 7a). However, at both Site 2 and Site 13, japonica and indica cultivars were clustered into two different clades (Supplementary Fig. 7b, c). Among the eight O. rufipogon accessions clustered together with cultivated rice at Site 7, seven were grouped into the adjacent clade with japonica at both Site 2 and Site 13 (Supplementary Fig. 7b, c). This indicated that although japonica and indica cultivars had different origins, the single deletion event responsible for plant architecture transition might have occurred during japonica rice domestication. Our study therefore provides additional evidence in support of the multiple origins, single domestication hypothesis with respect to the origin of Asian cultivated rice[21,24,25].

**Variation at the RPAD locus of Asian AA-genome wild rice.** PROG1 (located 3292-bp upstream from the deletion site) and three functional ZnFs (ZnF5, ZnF7, and ZnF8) within the SPROG1 locus all belong to the same gene family. Therefore, this locus comprises a gene cluster controlling plant architecture. We labeled this locus as the RPAD locus. We investigated the natural variations at the RPAD locus among wild rice by genotyping the deletion using PCR. All 39 accessions of Asian AA-genome wild rice from nine countries, including 35 accessions of O. rufipogon and four accessions of O. nivara, had amplicons similar to those of DXCWR, indicating that the wild rice all carried a large chromosomal segment at the RPAD locus (Supplementary Fig. 8).

We further assembled the full RPAD regions from two Asian AA-genome wild rice accessions; one was the O. rufipogon accession YJCWR from China used in identifying PROG1[18], and the other was an O. nivara accession W2014 from India. The RPAD locus in YJCWR and W2014 contains 143 and 110-kb insertion segments, and harbors a tandem repeat of eight and seven putative zinc-finger proteins, respectively (Fig. 5 and Supplementary Data 1). We also found several large segment insertions/deletions (indels) at this locus among DXCWR, YJCWR, and W2014 (Fig. 5).

To investigate the functional similarity of the alleles of zinc-finger gene among wild rice, we introduced a construct (CP-ZnF8$^{YJCWR}$) harboring the eighth zinc-finger gene from YJCWR, which differs in two amino acids with ZnF8 in DXCWR, into japonica variety ZH17. The results showed that the ZnF8 gene in YJCWR had the same role in controlling semi-prostrate growth (Supplementary Fig. 9) ($P < 0.01$, two-tailed $t$ test), indicating that the alleles of zinc-finger gene among wild rice at the RPAD locus might have similar function in regulating plant architecture development. Notably, we found that, in both YJCWR and W2014, the conserved $C_2H_2$ zinc-finger domain of the genes (gene ID: YJ_5 and NIV_7) corresponding to ZnF3 in DXCWR is altered because of DNA polymorphisms (Supplementary Fig. 10). Altogether, the variation of sequence and structure at the RPAD locus might be associated with the selective adaptation of growth habit in diverse O. rufipogon and O. nivara species, which will be valuable for further exploration of the underlying evolutionary mechanisms on growth habits in Asian AA-genome wild rice.

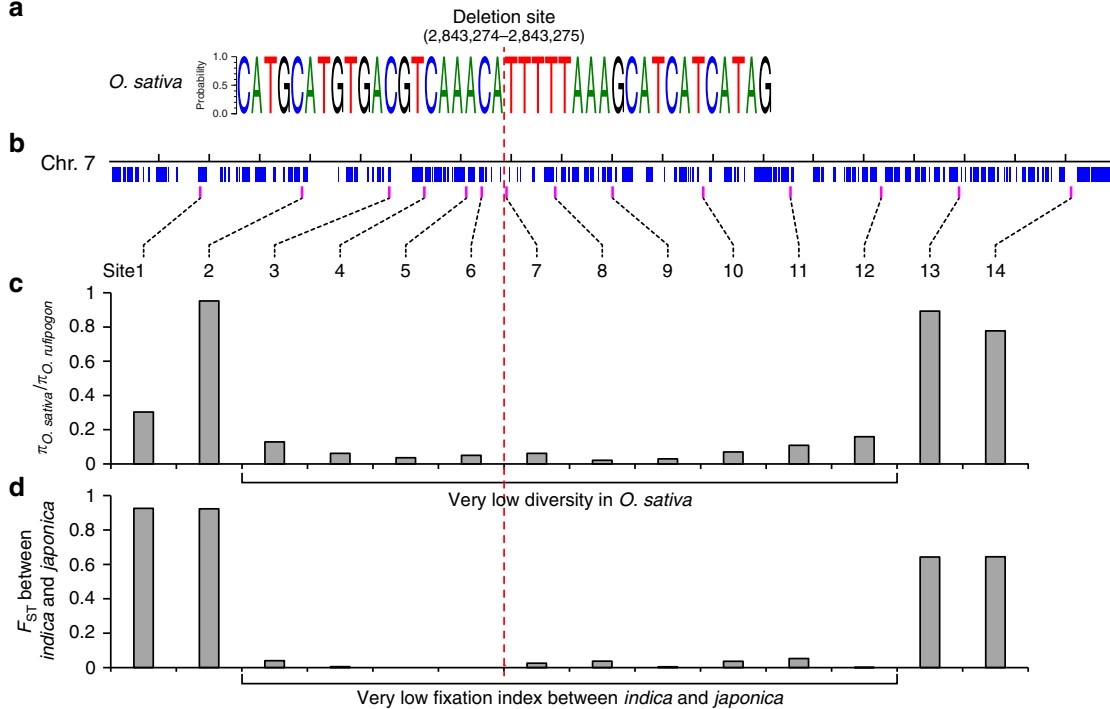

**Fig. 4** Nucleotide diversity and fixation index across the deletion site on chromosome 7. **a** The consensus sequence covering the breakpoint of deletion in 133 Asian rice varieties. All surveyed Asian rice cultivars had an identical deletion. The breakpoint of deletion located between 2,843,274 and 2,843,275 bp on chromosome 7 in the Nipponbare reference genome (Os-Nipponbare-Reference-IRGSP-1.0, MSU7). **b** The location of 14 sampled loci within the ~1 Mb genomic region surrounding the deletion site on chromosome 7. The position of each loci was showed in Supplementary Data 5. **c** The relative ratio of nucleotide diversity ($\pi$) in *O. sativa* to *O. rufipogon* shows the ~500 kb selective sweep surrounding the deletion site in Asian cultivated rice. **d** The fixation index ($F_{ST}$) between *indica* and *japonica* cultivars at the 14 sampled loci. The dashed red line indicates the deletion site in the genome of *O. sativa*

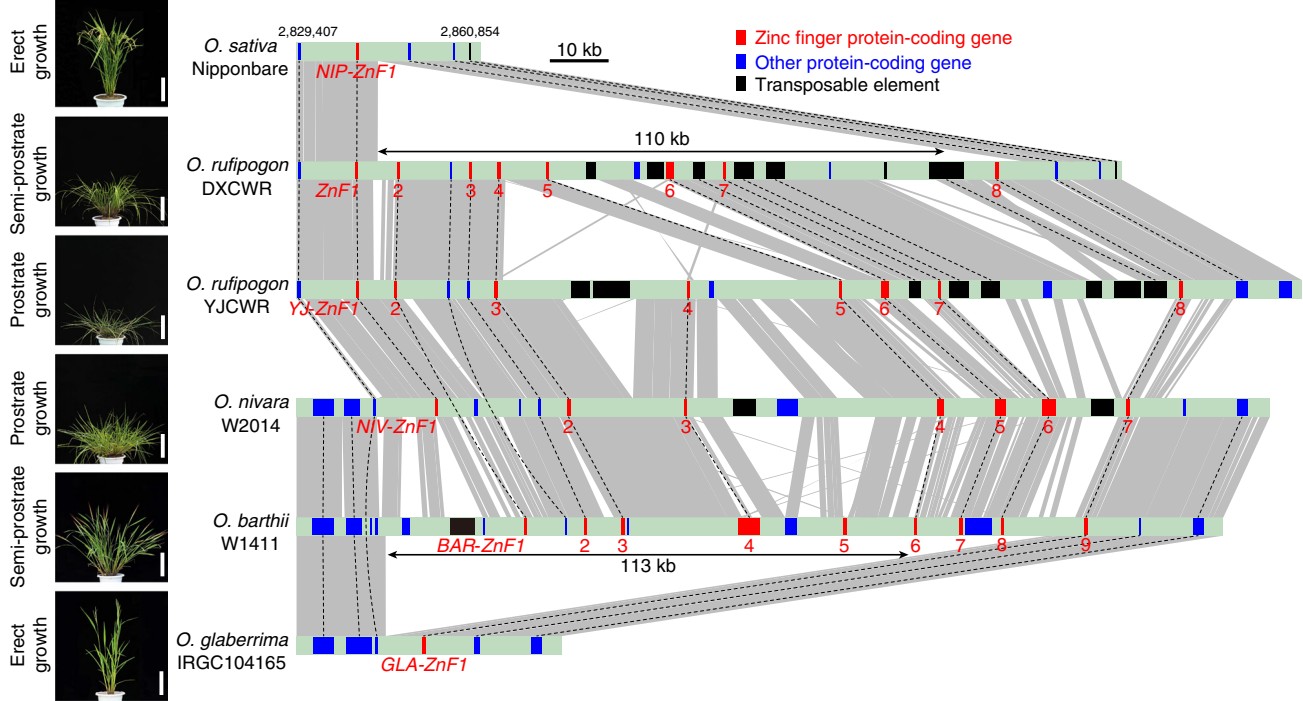

**Fig. 5** Structural variations at the *RPAD* locus in six *Oryza* AA genomes. The numbers (2,829,407 and 2,860,854) showed the location of the 5′ and 3′ end in the Nipponbare reference genome (Os-Nipponbare-Reference-IRGSP-1.0, MSU7). The red, blue, and black boxes respectively indicate zinc-finger protein-coding genes, other protein-coding genes, and transposable elements. The gray region represents those sharing sequence collinearity. Dashed black lines indicate the orthologous relationships of genes and transposable elements. *ZnF* genes were named based on the 5′–3′ order at the *RPAD* locus in the corresponding accession (Supplementary Data 1 and 3). Scale bars, 20 cm

**A convergent deletion is selected in African cultivated rice.** African cultivated rice (*O. glaberrima*) was domesticated around 3000 years ago in West Africa[26,27], which also underwent the similar transition from semi-prostrate to erect growth (Fig. 5). We performed QTL analysis using a $F_2$ population derived from a cross between *O. glaberrima* var. IRGC104165 and accession W1411 of *O. barthii*, the wild ancestor of *O. glaberrima*. We again detected a QTL associated with the transition of plant architecture localized at the *RPAD* locus on chromosome 7 (Supplementary Fig. 11a, b), indicating that a parallel selection mechanism might occur at the *RPAD* locus during the domestication of *O. glaberrima*.

Next, we assembled the genomic sequences in this locus from both parents and found that the genome of *O. glaberrima* contained a chromosomal deletion (113-kb) resulting in the absence of eight zinc-finger protein-coding genes (Fig. 5, Supplementary Fig. 11c–e and Supplementary Data 3). We further confirmed that all ten *O. barthii* accessions had amplicons similar to that in W1411, indicating that *O. barthii* genome has a large chromosomal segment within the *RPAD* locus. Meanwhile all 50 *O. glaberrima* varieties from eight countries of West Africa had the identical chromosomal deletion, indicating that the deletion might be fixed during African rice domestication (Supplementary Fig. 11c, d).

Surprisingly, the deletion sites are different between Asian and African cultivated rice. *O. glaberrima* maintained one zinc-finger gene (*GLA-ZnF1*), which is the ortholog of *BAR-ZnF9* in *O. barthii* and *ZnF8* in DXCWR (Fig. 5). We analyzed the expression level of the genes within the *RPAD* locus between *O. barthii* W1411 and *O. glaberrima* IRGC104165 by RNA-seq. *GLA-ZnF1* only expressed at a very low level in tiller base and not in other seven tissues of *O. glaberrima* IRGC104165. Meanwhile, the *BAR-ZnF5*, *BAR-ZnF7*, and *BAR-ZnF8* (the ortholog of the functional *ZnF7* in DXCWR) all had high expression level in tiller base of *O. barthii* W1411 (Supplementary Fig. 12), suggesting that the three *ZnF* genes might play important roles in the regulation of plant architecture in *O. barthii*.

Therefore, our findings implied that the selection and fixation of this deletion at the *RPAD* locus, coupled with the loss of *ZnF* genes, might be associated with the transition of plant architecture during the domestication of African cultivated rice. A commonly shared genetic modification might explain the convergent evolution of plant architecture in the two independently domesticated rice species at Asia and Africa.

## Discussion

Obtaining a plant architecture that maximizes yield is key to progressive crop domestication. Previous studies demonstrated that the *PROG1* gene is associated with plant architecture domestication in Asian cultivated rice and found that the expression level of *PROG1* was higher in wild rice than that in cultivated rice[17,18], indicating that the *PROG1* expression might also be altered during domestication. In the present study, the *SPROG1* fine-mapped region covered a part of 5′-flanking region of *PROG1* gene, in which there were two single nucleotide polymorphisms (SNPs) and two indels between GC2 and DXCWR (Supplementary Fig. 13), and *PROG1* is located at 3295-bp upstream of the deletion site. Therefore, it remains unclear whether the mutations within *PROG1* promoter or the deletion alter the expression of *PROG1*. Additionally, the 500-kb selective sweep spans both the *PROG1* gene and the deletion site, and Asian cultivated rice have the same *prog1* allele[18] and identical deletion site at the *RPAD* locus, indicating that the deletion is in linkage disequilibrium with the *prog1* allele. Hence, whether *PROG1* or the deletion at the *RPAD* locus were the targets of

selection during domestication remains to be investigated for better understanding the roles of these two loci in rice plant architecture domestication.

During crop domestication, humans modified wild plant species by selecting favorable genetic variations in order to improve their productivity and nutrition[1–3]. Previous works have demonstrated that causative mutations relating to crop domestication, including SNPs[4–6], small indels[7,11,13,14], and the presence or absence of mobile DNA elements[28], resulted in the dysfunction and/or alteration in the expression pattern of domestication-related genes. We found that the deletion (>100 kb deletion) within the *RPAD* locus associated with the parallel domestication of plant architecture in both Asian and African rice, indicating that structural variation played important roles during crop domestication. Notably, the collinear chromosomal region of the *RPAD* locus in foxtail millet (*Setaria italic*) and its presumed progenitor green foxtail (*S. viridis*) had a similar tandem repeat of zinc-finger protein-coding genes and harbored a complex structural variation (Supplementary Fig. 14 and Supplementary Table 3). This suggests that the *RPAD* locus might be recognized as an ancient zinc-finger gene cluster with a conserved functional role in the regulation of plant growth habit in the family Poaceae. Thus, the *RPAD* syntenic locus in other crops would be a strong candidate for improving of plant architecture and enhancing grain yield by genetic modification in the future.

## Methods

**Plant materials and growth conditions.** The introgression line DIL29 was derived from a cross between the recipient parent Guichao 2 (GC2, *O. sativa* ssp. *indica*) and the donor parent DXCWR, an *O. rufipogon* accession from the Dongxiang county of Jiangxi Province, China. Two mapping populations were developed, from a cross between introgression line DIL29 and GC2 and from a cross between African cultivated variety (IRGC104165, *O. glaberrima*) and an *O. barthii* accession (W1411). These plant materials were grown in the field at experimental stations of China Agriculture University in Beijing and Hainan, China. Information of 133 varieties of *O. sativa*, 35 accessions of *O. rufipogon*, four accessions of *O. nivara*, 50 varieties of *O. glaberrima*, and 14 accessions of *O. barthii* used in this study are listed in Supplementary Data 4.

**Primers.** The primers used in this study are listed in Supplementary Data 5.

**Phenotypic evaluation.** For the recipient parent Guichao 2, introgression line DIL29, and each homozygous recombinant used in the fine-mapping experiment, we used 10 plants to measure the tiller angle, tiller number, plant height, number of primary branches, secondary branches, grains on the main panicle, and grain yield per plant. For the transgenic plants, phenotypic measurements were performed using three independent transgenic lines (ten positive plants from each line).

**QTL mapping.** QTL analysis was performed by composite interval mapping with QTL IciMapping V4.0[29].

**Genome sequencing and de novo assembly of the *RPAD* locus.** For DXCWR (*O. rufipogon*) and YJCWR (*O. rufipogon*), the BAC clones were screened from the corresponding BAC library[30] and sequenced by shotgun sequencing. The BAC sequences were assembled using Lasergene version 14 (http://www.dnastar.com/t-allproducts.aspx). For W2014 (*O. nivara*), IRGC104165 (*O. glaberrima*), and W1411 (*O. barthii*), high-molecular-weight genomic DNA was extracted from a single seedling leaf using a DNeasy Plant Mini Kit (QIAGEN, Germany). Libraries were prepared and sequenced on the platforms of PacBio single molecule, real-time (SMRT) RS II (~25× coverage) and Illumina HiSeq2500 (~50× coverage) at the BerryGenomics Company (China). The draft genome sequence of the syntenic region was first assembled with a genome walking approach using Lasergene by extracting PacBio reads. Then, the genome sequence of the syntenic region was reassembled and corrected using the genome draft-related reads from PacBio and Illumina reads by DBG2OLC[31]. The putative genes in the *RPAD* locus were predicted using FGENESH[32] and annotated by Blast2GO version 4.1[33], with minor modification.

**Vector construction and rice transformation.** The 11 fragments from the DXCWR genome in the fine-mapped region of *SPROG1* were obtained using either PCR amplification or an ultrasonication approach and inserted into the binary vector pCAMBIA1300 to generate the complementary constructs (Supplementary

Table 2). To investigate the function of $ZnF8^{YJCWR}$ from YJCWR (the allele of $ZnF8$ in DXCWR), the 3487-bp fragment harboring the entire $ZnF8^{YJCWR}$ sequence of YJCWR was inserted into pCAMBIA1300 to generate the CP-$ZnF8^{YJCWR}$ construct. All plasmid constructs and corresponding empty vectors were introduced into *Agrobacterium tumefaciens* strain EHA105 and subsequently transferred into the *japonica* cultivar Zhonghua 17 (ZH17).

**Subcellular localization**. The coding sequences of $ZnF5$, $ZnF7$, and $ZnF8$ were cloned into p2GWF7[34] with Gateway cloning technology (Invitrogen, USA) to generate the ZnF5-, ZnF7-, and ZnF8-GFP constructs, respectively. Similarly, the coding sequence of $OsMADS15$[35] was cloned into p2GWR7 to generate OsMADS15-RFP as a nuclear marker. The ZnF5-, ZnF7-, or ZnF8-GFP and OsMADS15-RFP constructs were co-transformed into rice protoplasts using the polyethylene glycol method[36]. After 20–24 h of incubation at 28 °C, GFP and RFP fluorescence were observed using a confocal laser-scanning microscope (Carl Zeiss LAM510, Germany).

**Transcriptional activity assay in rice protoplasts**. The reporter plasmid GAL4-LUC includes the firefly *luciferase* gene driven by the minimal TATA box of the *CaMV 35s* promoter following five repeats of the yeast GAL4 protein-binding element[37]. The effector plasmids were constructed by fusing the entire or 3′ truncated (EAR motif deletion) coding region of $ZnF5$, $ZnF7$, and $ZnF8$ with GAL4-BD, respectively. Plasmid pRTL, which included the *Renilla luciferase* gene driven by *CaMV 35s*, was used as the internal control for normalization. For each assay, 6 μg of reporter plasmid DNA, 6 μg of effector plasmid DNA, and 1 μg of internal control plasmid DNA were co-transformed into rice protoplasts with the polyethylene glycol-mediated method[36]. After incubating for 16–24 h at 28 °C, the relative luciferase activities were measured using the Dual-Luciferase Reporter Assay System (Promega, USA) following the manufacturer's instructions.

**RNA-seq data analysis**. For transcriptome analysis, total RNA of ten tissues from DXCWR plants, eight tissues from IRGC104165 plants, and eight tissues from W1411 plants was respectively isolated using TRIzol (Invitrogen, USA) according to the instructions. Paired-end libraries were constructed using a TruSeq RNA Sample Preparation Kit v2 (Illumina, USA) according to the manufacturer's instructions and were sequenced with an Illumina system Hiseq2500 at the BerryGenomics Company (China). RNA-seq reads were aligned to the genome sequence of the collinear region in DXCWR using TopHat[38]. Only uniquely mapped reads were selected and then visualized on an Integrated Genomics Viewer[39]. To discover the genes regulated by functional $ZnF$ genes within the $RPAD$ locus, total RNA was isolated from tiller bases of the CP-PROG1, CP-ZnF5, CP-ZnF7, CP-ZnF8 transgenic lines and from control plants (ZH17) at 50 days after sowing using three biological replicates. Paired-end libraries were constructed and sequenced with an Illumina system Hiseq2500 at the BerryGenomics Company (China). The reads were mapped to the reference genome (Os-Nipponbare-Reference-IRGSP-1.0, MSU7) by TopHat[38] using default parameters. Cuffdiff[40] was used to calculate the fragments per kilobase of exon per million mapped reads (FPKM) of each gene and to identify the differentially expressed genes (fold change ≥ 2, $P < 0.001$, negative binomial test, and FPKM value in each sample is greater than 1) between the each transgenic line (CP-PROG1, CP-ZnF5, CP-ZnF7, or CP-ZnF8) and the control. The functional category analysis of the differentially expressed genes was performed by the agriGO[41] and KEGG[42].

**RNA in situ hybridization**. After 30 days of sowing the tiller bases of introgression line DIL29, they were fixed in 3.7% (v/v) FAA solution, dehydrated, embedded in paraffin (Thermo Fisher Scientific, USA), and sliced into 8-μm sections with a microtome (Leica RM2145, Germany). Three fragments of $ZnF5$, $ZnF7$, and $ZnF8$ cDNA from DIL29 was amplified, respectively and used as the template to generate sense and antisense RNA probes. Digoxigenin-labeled RNA probes were prepared using a DIG RNA labeling kit (Roche, Switzerland). RNA hybridization and immunological detection of the hybridized probes were performed following the methods of Zhang et al.[43].

**Selective sweep analysis**. Multiple sequences were aligned using the ClustalW program in MEGA version 6.0[44]. The average proportion of pairwise differences per base pair ($\pi$), fixation index ($F_{ST}$), and Tajima's $D$ test were calculated using DnaSP version 5.1[45].

**Collinearity analysis**. MCScanX version 0.8[46] was used to identify the collinearity genes through generating dot plot alignments between rice and *S. italica* with default parameters. The genomic collinearity map of the $RPAD$ locus was plotted according to the results of BLASTN[47].

**Statistical analysis**. Two-tailed Student's *t* test was used with SPSS version 16 (SPSS Inc., Chicago). Statistical significance was set at $P < 0.01$.

## Data availability

The GenBank accessions for nucleotide sequences of $RPAD$ locus from Guichao 2, DXCWR, YJCWR, W2014, IRGC104165, and W1411 are MF503969, MF503970, MF503971, MF503972, MF503973, and MF503974, respectively. Short-read data generated in this study have been deposited in NCBI's Short-Read Archive under the accession SRP150515. RNA-seq reads of ten tissues from DXCWR, eight tissues from IRGC104165, and eight tissues from W1411 have been deposited in NCBI under the accession SRP151515. RNA-seq data of transgenic plants have been deposited in NCBI's Gene Expression Omnibus under the accession GSE116423. The GenBank accessions for nucleotide sequences of Sites 2, 13, and 7 are MH293609 to MH294118. Data that support the findings of this study are available from the corresponding author upon reasonable request.

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

## Acknowledgments

We thank Prof. Xianmin Diao of Chinese Academy of Agricultural Sciences for sharing foxtail millet and green millet germplasm, Prof. Shou-Yi Chen of Chinese Academy of Sciences for providing the DLR assay system, Dr. Roger Y. Tsien of University of California, San Diego, for providing the p2GWR7 vector, and International Rice Research Institute, United States Department of Agriculture-Agricultural Research Service, National Institute of Genetics (Japan), Institute of Crop Sciences of Chinese Academy of Agricultural Sciences for providing the wild rice germplasm. This work was supported by National Key Research and Development Program of China (Grant no. 2016YFD0100400), National Natural Science Foundation of China (Grant nos. 91435103 and 31222040), and Chinese Universities Scientific Fund (Grant no. 2017NX002).

## Author contributions

L.T. designed the experiments and managed the project. Y.W., S.Z. and X.L. performed most of the experiments. B.Z., L.J., Y.T., J.Z., X.M., H.C. and C.S. provided technical assistance and conducted the collection and maintenance of rice germplasm. L.T., Y.W., X.L. and S.Z. performed data analysis and wrote the manuscript.
