## [Peer Review File · Nature Communications]

Reviewers' comments:

Reviewer #1 (Remarks to the Author):

In the study, Wu et al reported the map-based cloning of a quantitative trait locus RPAD (RICE PLANT ARCHITECTURE DOMESTICATION) for plant architecture and its involvement of rice domestication. Sequence variation of parental alleles at RPAD points to the deletion of large fragment (~110 kb) of chromosome in cultivated Asian rice varieties; however, the corresponding segment of wild rice contains a cluster of ZnF genes, some of which were confirmed to function in plant architecture regulation. It was interesting that an equivalent fragment deletion of RPAD alleles occurred in African cultivated rice varieties compared to that of corresponding wild African rice. Together, these findings added an important contribution to improve our understanding of crop domestication in plant architecture. That being said, I still have my main concern about what were the possible molecular, physiological and cytological mechanisms of the RPAD regulation of plant architecture.

Also, I was curious that is there any functional redundancy among these functional ZnF genes in plant architecture regulation? Especially, I noticed that ZnF8 gene was present at RAPD in *O. glaberrima* IRGC104165; however, its plants grow erectly (Figure 5). Page 8, Line 231, what does the "BAR-ZnF9" means (the authors were suggested to mark the gene correspondingly in Figure 5)? Also in Figure 5, might the *O. rufipogon* and *O. nivara* ZnF3 alleles were marked incorrectly?

Furthermore, the authors were suggested to summarize the novelty of RPAD regulation of plant architecture that is distinct from that of the reported PROG1 gene in the discussion.

The manuscript could benefit from further English editing.

Reviewer #2 (Remarks to the Author):

This paper by Wu et al. identifies an ~110 Kb deletion on chromosome 7 that seems to be responsible for the erect growth and high-yield phenotype of cultivated rice in comparison to the more prostrate growth of wild rice. This transition in plant architecture was one of great consequence during rice domestication, and the functional evidence shown for a role of this deletion in the phenotype is compelling. That a similar deletion is then identified by the authors to have occurred in the transition from *O. barthii* to *O. glaberrima* is surprising and very exciting. Thus, I think this study is an important contribution to our understanding of how rice has been domesticated.

I'd like to start off by saying that I am puzzled by the term "microdeletion"; am I right in suspecting this comes from the genetic disease field? I can certainly think of several much smaller deletions in other genes that have been of importance during rice domestication.

A curious aspect of these results is that another locus on this chromosome, known as PROG1, had previously been identified as being involved in the transition to erect growth during rice domestication. Because the deletion site and PROG1 are only ~3000 bp apart, the loci are linked. I was somewhat frustrated that the authors did not discuss any implications of their finding on the possible role of PROG1. My impression is that causal polymorphisms in PROG1 have not been identified, and that variants at this locus do not map to predicted phenotypes as well as one would wish. Does the discovery of this deletion imply that there is no role for PROG1 in rice domestication, and that the only functional polymorphism in the region is the presence/absence of the deletion?

I appreciate the population level analyses, which looked at the presence of the deletion across

numerous samples of wild and cultivated rice and examined the possibility of sweeps. However, I feel that the authors have been very cursory in much of the accessory data provided here and elsewhere. There is no information in the supplementary data of what germplasm repository rice samples were obtained from, and the lab ID provided is not sufficient for readers who might want to work with the same samples (names are often quite malleable for crop germplasm; the germplasm repository IDs are much more useful). There is no information on the rice genome version used in localizing the RPAD locus, nor are positions on the reference genome provided for the primers used in selective sweeps. For the initial mapping of SPROG1, no information is provided on the type of markers used (there are primer sequences, but are the markers based on SNPs?) or their positions in the reference genome. For the phylogenetic analyses, there is no explanation of what size DNA fragments were used at sites 2, 13, and 7. I am also bemused by the data availability statement provided; who decides what is a reasonable request?

The argument that the results shown here suggest a single origin of the 110 Kb deletion in japonica, and then a transfer of this allele to early domesticated indicas is persuasive. However, I wonder how well this argument holds up in light of the recent evidence from Civan et al (Genet Resour Crop Evol) showing that many of the identified rice domestication alleles at other loci segregate in *O. rufipogon* populations and can thus have been independently derived in indica and japonica. That the *O. rufipogon* samples used here seem somewhat biased towards China increases the worry that these may not represent the full diversity of the RPAD locus within the wild species.

Reviewer #3 comments are attached.

[Editorial Note: Please see Reviewer #3 comments on the following page.]

Wu et al. provide an excellent report of the role of deletions at RPAD in the domestication of Asian and African rice. This is a very nice study representing a ton of work and careful analysis. I nonetheless have some comments that I think would improve the manuscript. I'd be happy to answer questions or clarify comments if I can be of additional help, and I apologize if I missed or misinterpreted anything.

Best,
Jeffrey Ross-Ibarra

Main comments

- I don't understand why the authors do not take advantage of previously sequenced rice genomes for their population genetic analysis. Hundreds of rufipogon and thousands of cultivated rice genomes have been sequenced and are publicly available. Many are low coverage, but existing genotype likelihood methods make analysis of such genomes feasible. Doing so would allow more powerful tests, comparison of RPAD to other genomic regions (does it show more or less selection than other domestication genes), and would also help when the authors seek to understand whether the deletion they identify is found in other lines. Since the breakpoints of the deletion are well known, simply aligning short read data from public sources should allow detection of presence/absence of the deletion in many additional lines (including wild taxa).
- I'm a bit skeptical of the phylogenetic analysis (line 158 and after). It doesn't add much, especially when conducted at so few sites. It's also not totally clear to me how the authors can detect the direction of gene flow from japonica into indica from this analysis. Perhaps a more careful introgression analysis is warranted (formal tests for introgression), or this analysis could be performed genome-wide for comparison. Or maybe some of this section could simply be dropped.
- In addition to the RPAD locus in Genbank, the authors need to make the short read data available on the SRA. I would also strongly encourage the authors to share other data forms from the paper publicly (via figshare or similar service). "available upon request" is problematic for reproducibility, even from the most well-intentioned authors. For example, I no longer have access to some code that former papers from my lab claim "is available upon request", meaning the code is no longer available to any reader. Putting code on github, data on figshare, etc. ensures the data will be there for future readers.

Minor comments

- I don't like the term microdeletion. 100kb is pretty darned big! Why not just say "deletion"? It is even called a "large genomic change" later (line 244).
- Line 62: what is the citation for identical mutations in prog1?
- line 154: I don't know what "no significant differentiation occurred in the sweeping selection between" means. I suspect this is just awkward phrasing?
- I think the PCR results reported on line 182 conflicts with the phylogeny for site 7 reported on 162. The former suggest all wild rice are similar, but the latter find 8 rufipogon that look like cultivated rice. Did I misunderstand something?
- line 352: should say "RPAD"
- line 205: I don't believe you can really call this a "hot spot" without a quantitative comparison to rates of insertion/deletion over the rest of the genome?
- line 175: is this really the right citation for the "multiple origins, single domestication" hypothesis? Isn't that basically one of the models Tao Sang and Song Ge came up with in 2007?

Dear Reviewers,

Thank you very much for your helpful comments on our manuscript "Chromosomal microdeletions drove parallel domestication of plant architecture in Asian and African rice" (NCOMMS-17-34535-T). We have carried out additional experiments, added two Supplementary Figures (Supplementary Figures 5 and 11) and one Supplementary Table (Supplementary Table 4), and revised the manuscript accordingly. We used red color to mark the modified words and sentences in main text and figure legends. The detailed responses are shown below. We would be grateful if you can kindly be satisfied with the revised manuscript.

Thank you again for your time and efforts in taking care of our manuscript.

Sincerely,

Lubin Tan, Ph.D

Department of Plant Genetics and Breeding

China Agricultural University, Beijing 100193, China

Responses to reviewer 1's comments:

In the study, Wu et al reported the map-based cloning of a quantitative trait locus RPAD (RICE PLANT ARCHITECTURE DOMESTICATION) for plant architecture and its involvement of rice domestication. Sequence variation of parental alleles at RPAD points to the deletion of large fragment (~110 kb) of chromosome in cultivated Asian rice varieties; however, the corresponding segment of wild rice contains a cluster of ZnF genes, some of which were confirmed to function in plant architecture regulation. It was interesting that an equivalent fragment deletion of RPAD alleles occurred in African cultivated rice varieties compared to that of corresponding wild African rice. Together, these findings added an important contribution to improve our understanding of crop domestication in plant architecture.

Response: Thank you very much for your positive comments and helpful suggestions.

Reviewer 1's question 1 (R1Q1): That being said, I still have my main concern about what were the possible molecular, physiological and cytological mechanisms of the RPAD regulation of plant architecture. Also, I was curious that is there any functional redundancy among these functional ZnF genes in plant architecture regulation?

Response: To explore the possible molecular mechanism of the *RPAD* regulation of plant architecture and the functional redundancy among these functional *ZnF* genes, we performed RNA-seq experiments using the transgenic plants of functional *ZnF* genes, including *ZnF5*, *ZnF7*, *ZnF8*, also *PROG1* (*YJ-ZnF1*) as well because of other concerns like R1Q5 and R2Q2. We identified a total of 4,431 genes have different expression levels (fold change ≥ 2 , $P < 0.001$) between at least one *ZnF* transgenic plant and the controls (ZH17) (new Supplementary Table 4). Cluster analysis of the 4,431 differentially expressed genes (DEGs) showed that the similarity is particularly high between *ZnF5* and *ZnF7* transgenic plants (new Supplementary Fig. 5a), which is in accordance with phylogenetic analysis of these ZnF protein sequences (Supplementary Fig. 3b). In addition, venn diagrams analysis showed that a substantial amount of DEGs (approximately 69.3%) were uniquely regulated by individual *ZnF* gene were discovered, and 85 DGEs were commonly regulated by four functional *ZnF* genes (new Supplementary Fig. 5b and c). Further gene ontology (GO) analysis of the DEGs commonly regulated by both *ZnF5* and *ZnF7* showed that these DEGs were enriched in multiple biological processes, including transcription regulator activity, biosynthesis of secondary metabolites, stimulus response, carotenoid biosynthesis, and plant hormone signal transduction (new Supplementary Fig. 5d), suggesting regulating plant architecture requires a complex network. With these new evidences from transcriptome profiling, these functional *ZnF* genes might regulate plant architecture through a shared pathway, but also have specific functions in regulating other pathways. Further detailed dissection for functional redundancies requires characterization of a series of mutants

knocking out each *ZnF* gene in wild rice background, which remains challenging because of the transformation for wild rice.

The result of transcriptome analysis have added in the third paragraph of page 5. We would be pleased if you can be satisfied with our results and explanation. Thank you!

RIQ2: Especially, I noticed that *ZnF8* gene was present at *RAPD* in *O. glaberrima* IRGC104165; however, its plants grow erectly (Figure 5).

Response: Yes, *O. glaberrima* cultivar IRGC104165 grows erectly. Although it carries one *ZnF* gene *GLA-ZnF1*, which is the ortholog of *ZnF8* of DXCWR (Please refer to response for RIQ3 about the nomenclature of these *ZnFs*), the *GLA-ZnF1* allele in *O. glaberrima* is a recessive allele because of the result of QTL mapping (Supplementary Fig. 10a,b). Comparison of amino acid sequences between *GLA-ZnF1* and *BAR-ZnF9* (the allele of *GLA-ZnF1* in *O. barthii* accession W1411) identified one single-amino acid deletion, one four-amino acid insertion, and two single-amino acid substitution mutations in *GLA-ZnF1* (Supplementary Fig. 10e and Response Fig. 1). We also found multiple mutations in 5'-flanking region of this gene between *O. glaberrima* and *O. barthii*. We further analyzed the expression level of the putative genes within the *RPAD* locus of *O. barthii* accession W1411 and *O. glaberrima* cultivar IRGC104165 by RNA-seq. The result showed that *GLA-ZnF1* only expressed at a very low level in tiller base and could not be detected in seven other tissues of *O. glaberrima* IRGC104165. Therefore, we speculated that the variations in either from coding region or 5'-flanking region might disrupt the *GLA-ZnF1* function and inactive *GLA-ZnF1* expression. Meanwhile, the *BAR-ZnF5*, *BAR-ZnF7*, and *BAR-ZnF8* (the ortholog of the functional *ZnF7* in DXCWR) all had high expression level in tiller base of *O. barthii* W1411 (new Supplementary Fig. 11), suggesting that the deletion in the *RPAD* locus, effacing a tandem repeat of zinc-finger genes, also plays an important role in the plant architecture domestication of African cultivated rice. The result of the expression level of the putative genes within the *RPAD* locus of *O. barthii* and *O. glaberrima* have added in the

third paragraph of page 8.

In this study, we focus on the comparison of the genome structure variation at the *RPAD* locus between Asian and African rice, and found that a similar but independent ~113-kb deletion event in African cultivated rice might be associated with the transition of plant architecture during the domestication of African cultivated rice. However, without evidences from further experiments, we cannot conclude which *ZnF* gene(s) controls for growth habit transition. We would be delightful if you can allow us to further investigate the function and causal variations of *ZnF* genes in the *RPAD* locus of *O. barthii* and *O. glaberrima* in our future project.

GLA-ZnF1	1	MEPCRR-HDGGGGGGRRSSRVFECLFCDKTFHKSQALGGHQNAHKKDHVAAAGDWDPPYYVHG	63
BAR-ZnF9	1	MEPCRRRHGTVGGGGGGRRSSRVFECLFCDKTFHKSQALGGHQNAHKKDHVAAAGDWDPPYYVHG	64
GLA-ZnF1	64	NGIHPAAAAATAARDPYAGYPAASTTMPLPPVAGGRTPHGAVVTAPGLVFAATSRPLRPLPH	127
BAR-ZnF9	65	NGIHPAAAAATAARDPYAGYPAASTTMPLPPVAGGRTPHGAVVTAPGLVFAATSRPLRPLPH	128
GLA-ZnF1	128	GHGVAAGSGGWHDIRAWPMEYSVDDGAASFFRASRKDGGDATVDDVVVDGGEVEVLDLELRL	191
BAR-ZnF9	129	GHGVAAGSGGWHDIRAWPMEYSVDDGAASFFGASRKDGGDAT----VVDGGEVEVLDLELRL	188

Response Fig. 1 Amino acid polymorphisms between GLA-ZnF1 and BAR-ZnF9.

RIQ3: Page 8, Line 231, what does the “BAR-ZnF9” means (the authors were suggested to mark the gene correspondingly in Figure 5)?

Response: We wished to have a better nomenclature system for these *ZnF* genes. However, the complex of *RPAD* locus made it is a challenging task. The current way names these *ZnF* genes based on the 5' to 3' end order at the *RPAD* locus in the corresponding accession (Supplementary Tables 2 and 5). For example, *GLA-ZnF1* is the first *ZnF* genes in *O. glaberrima*, and *BAR-ZNF9* is the ninth annotated *ZnF* genes in *O. barthii*, but *BAR-ZnF9* is the allele of *GLA-ZnF1*, and both are the ortholog of *ZnF8* in *O. rufipogon* DXCWR. According to your suggestion, we have marked each *ZnF* genes in Figure 5 and added an addition column in Supplementary Tables 2 and 5 to highlight the ortholog in *O. rufipogon* DXCWR.

RIQ4: Also in Figure 5, might the *O. rufipogon* and *O. nivara* ZnF3 alleles were marked incorrectly?

Response: Based on the results of BLASTP, we considered that both the YJ-5 gene of *O. rufipogon* accession YJCWR and NIV-7 gene of *O. nivara* accession W2014 were the allele of *ZnF3* of *O. rufipogon* accession DXCWR. However, due to the genome variation, the predicated proteins of both the YJ-5 gene of *O. rufipogon* YJCWR and NIV-7 gene of *O. nivara* W2014 had no C₂H₂ zinc-finger domain (Supplementary Figure 9). Thus, both the YJ-5 gene of *O. rufipogon* YJCWR and NIV-7 gene of *O. nivara* W2014 were annotated as “other protein-coding gene” and marked in blue.

RIQ5: Furthermore, the authors were suggested to summarize the novelty of RPAD regulation of plant architecture that is distinct from that of the reported PROG1 gene in the discussion.

Response: According to your suggestion, we performed RNA-seq experiments using the transgenic plants of four functional *ZnF* genes at the *RPAD* locus. Please refer to response for R1Q3 about the results of transcriptome profiling. We have added the results in our revised manuscript.

RIQ6: The manuscript could benefit from further English editing.

Response: We have tried our best to improve our written English. Thank you.

Responses to reviewer 2's comments:

This paper by Wu et al. identifies an ~110 Kb deletion on chromosome 7 that seems to be responsible for the erect growth and high-yield phenotype of cultivated rice in comparison to the more prostrate growth of wild rice. This transition in plant architecture was one of great consequence during rice domestication, and the functional evidence shown for a role of this deletion in the phenotype is compelling. That a similar deletion is then identified by the authors to have occurred in the transition from *O.*

barthii to *O. glaberrima* is surprising and very exciting. Thus, I think this study is an important contribution to our understanding of how rice has been domesticated.

Response: Thank you very much for your positive comments.

R2Q1: I'd like to start off by saying that I am puzzled by the term "microdeletion"; am I right in suspecting this comes from the genetic disease field? I can certainly think of several much smaller deletions in other genes that have been of importance during rice domestication.

Response: According to your and Reviewer 3's suggestion, we have revised "microdeletion" to "deletion" throughout the text.

R2Q2: A curious aspect of these results is that another locus on this chromosome, known as *PROG1*, had previously been identified as being involved in the transition to erect growth during rice domestication. Because the deletion site and *PROG1* are only ~3000 bp apart, the loci are linked. I was somewhat frustrated that the authors did not discuss any implications of their finding on the possible role of *PROG1*. My impression is that causal polymorphisms in *PROG1* have not been identified, and that variants at this locus do not map to predict phenotypes as well as one would wish. Does the discovery of this deletion imply that there is no role for *PROG1* in rice domestication, and that the only functional polymorphism in the region is the presence/absence of the deletion?

Response: Thank you for your helpful comment. Based on two previous reports on *PROG1* (Tan et al. *Nature Genetics*, 40:1360–1364 and Jin et al. *Nature Genetics*, 40:1365–1369), the expression level of *PROG1* gene, which is encoded in the reverse strand, was higher in wild rice than that in cultivated rice (an *indica* variety Teqing), indicating that the change of *PROG1* expression might associate with plant architecture domestication in rice. In the current study, the fine-mapping region of QTL controlling plant architecture covered a part of 5'-flanking region of *PROG1* gene, in which there

were four SNPs and three indels between GC2 and DXCWR. In addition, *PROG1* is located 3,292 bp upstream from the 110-kb deletion site. Therefore, we speculate that the mutations in 5'-flanking region of *PROG1* gene and the 110-kb deletion may affect *PROG1* expression. However, we didn't identify the causal mutation(s) for the change of *PROG1* expression based on our current results. Meanwhile, whether the polymorphisms within *PROG1* and its promoter was the consequence of the deletion or occurred the same time needs further investigations. We would be delightful if you can allow us to further analysis the key mutations of *PROG1* in our future project.

R2Q3: I appreciate the population level analyses, which looked at the presence of the deletion across numerous samples of wild and cultivated rice and examined the possibility of sweeps. However, I feel that the authors have been very cursory in much of the accessory data provided here and elsewhere. There is no information in the supplementary data of what germplasm repository rice samples were obtained from, and the lab ID provided is not sufficient for readers who might want to work with the same samples (names are often quite malleable for crop germplasm; the germplasm repository IDs are much more useful).

Response: According to your suggestion, we have added the sample information in the supplementary Table 7. All rice samples collected by our institute (National Center for Evaluation of Agricultural Wild Plants) are available via material transfer agreement for research purposes. Thank you.

R2Q4: There is no information on the rice genome version used in localizing the RPAD locus, nor are positions on the reference genome provided for the primers used in selective sweeps.

Response: We have added a description of the reference genome (Os-Nipponbare-Reference-IRGSP-1.0, MSU7) in the legend of Fig. 4 and provided the positions on the reference genome of the primers in Supplementary Table 8.

R2Q5: For the initial mapping of SPROG1, no information is provided on the type of markers used (there are primer sequences, but are the markers based on SNPs?) or their positions in the reference genome.

Response: We have added the type of markers and the position of amplification fragment in Supplementary Table 8. Thank you.

R2Q6: For the phylogenetic analyses, there is no explanation of what size DNA fragments were used at sites 2, 13, and 7. I am also bemused by the data availability statement provided; who decides what is a reasonable request?

Response: According to your suggestion, we have added the position of amplification fragment of Sites 2, 7, and 13 in Supplementary Table 8, and deposited the sequences of Sites 2, 7, and 13 to GenBank under the accessions (MH293609–MH294118). We also have deposited other sequence data generated in this study, including short read data and RNA-seq data, in public database and revised the data availability statement in our manuscript.

R2Q7: The argument that the results shown here suggest a single origin of the 110 Kb deletion in japonica, and then a transfer of this allele to early domesticated indicas is persuasive. However, I wonder how well this argument holds up in light of the recent evidence from Civan et al (Genet Resour Crop Evol) showing that many of the identified rice domestication alleles at other loci segregate in *O. rufipogon* populations and can thus have been independently derived in indica and japonica. That the *O. rufipogon* samples used here seem somewhat biased towards China increases the worry that these may not represent the full diversity of the RPAD locus within the wild species.

Response: Thank you very much for your comment. According to your and Reviewer 3's suggestion, we had removed the speculation about genome introgression in our

revised manuscript.

Responses to reviewer 3's comments:

Wu et al. provide an excellent report of the role of deletions at RPAD in the domestication of Asian and African rice. This is a very nice study representing a ton of work and careful analysis. I nonetheless have some comments that I think would improve the manuscript. I'd be happy to answer questions or clarify comments if I can be of additional help, and I apologize if I missed or misinterpreted anything.

Response: Thank you very much for your positive comments and helpful suggestions.

R3Q1: I don't understand why the authors do not take advantage of previously sequenced rice genomes for their population genetic analysis. Hundreds of rufipogon and thousands of cultivated rice genomes have been sequenced and are publicly available. Many are low coverage, but existing genotype likelihood methods make analysis of such genomes feasible. Doing so would allow more powerful tests, comparison of RPAD to other genomic regions (does it show more or less selection than other domestication genes), and would also help when the authors seek to understand whether the deletion they identify is found in other lines. Since the breakpoints of the deletion are well known, simply aligning short read data from public sources should allow detecting of presence/absence of the deletion in many additional lines (including wild taxa).

Response: Because *PROG1* at the *RPAD* locus is a known gene for plant architecture domestication in rice, several previous studies reported that a strong selection signal was detected at the region embedding *PROG1* using genome-wide SNP data from lots of wild rice and cultivated rice (Xu et al., Nature Biotechnology, 30:105–111; Huang et al., Nature, 490:497–501). Therefore, in this study, we focused on the detection of deletion at the *RPAD* locus and found that all surveyed cultivars had an identical deletion, indicating that the deletion was fixed and strongly selected during rice

domestication. In addition, we detected a 500-kb selective sweep with a dramatic reduction of relative nucleotide diversity in the comparison of cultivated and wild rice, consistent with the previous report (Huang et al., Nature, 490:497–501).

According to your suggestion, we downloaded the genome resequencing data of 1,082 accessions of cultivated rice from the previous report (Huang et al., Nature, 490:497–501). These reads were firstly aligned to the 600-bp segment surrounding the breakpoint of the deletion with 300-bp from each side. Because the read depth of these 1,082 accession ranges from 0.3X to 2.6X (with the mean of 0.96x), the chance of recovering reads mapped to 600-bp from a single accession is low. So, we bin the accessions with sequence depth and recovered 717 accessions having reads (paired reads spanning the breakpoint of deletion and/or a read targeting the breakpoint of deletion) directly supporting the deletion event. We found that, among 67 accessions with >1.5X read depth, 57 (approximately 85%) had read(s) supporting the deletion event (Response Fig. 2a). The results indicated that as the sequence depth increase, the chance of recovering reads directly aligned to the deletion site increases. Moreover, the consensus sequence of all reads covering the deletion site from 415 accessions, including 193 *indica*, 8 *aus*, 176 *temperate japonica*, 16 *tropical japonica*, 4 *aromatic*, and 18 intermedia ecotype cultivars, is the same as we reported in current MS (Response Fig. 2b), indicating that Asian cultivated rice had an identical deletion within the *RPAD* locus. Therefore, these public available sources indeed offer extra evidence from large samples supporting our conclusion.

Asian wild rice is heavily admixed with domesticated rice genomes, as indicated that 26% of wild rice samples carrying domesticated *prog1* alleles (Genome Research, 27:1029–1038). In addition, we feel it is necessary to know the explicit information about the prostrate or erect growth of wild rice accessions. So, it may be difficult to truly reflect the characteristics of wild rice (the 110-kb deletion) when we conducted similar analysis with short reads from wild rice accessions. We would be delightful if you can allow us not to mention these data in our revised manuscript.

Response Fig. 2 Analysis of the deletion event at the *RPAD* locus using short reads from 1,082 resequenced cultivar rice accessions provided by Huang et al. (Nature, 490:497–501). **a** The chance of recovering reads, including paired reads spanning the breakpoint of deletion and/or a read targeting the breakpoint of deletion at the *RPAD* locus, directly supported the deletion event increases as sequence depth increase. The proportion indicates the ratio of the number of accession having read(s) supporting the deletion event to total number of accession. **b** The consensus sequence of all reads covering the deletion site from 415 accessions.

R3Q2: I'm a bit skeptical of the phylogenetic analysis (line 158 and after). It doesn't add much, especially when conducted at so few sites. It's also not totally clear to me how the authors can detect the direction of gene flow from japonica into indica from this analysis. Perhaps a more careful introgression analysis is warranted (formal tests for introgression), or this analysis could be performed genome-wide for comparison. Or maybe some of this section could simply be dropped.

Response: Thank you very much for your comment. According to your and Reviewer 2's suggestion, we had removed the speculation about genome introgression in our

revised manuscript.

R3Q3: In addition to the RPAD locus in Genbank, the authors need to make the short read data available on the SRA. I would also strongly encourage the authors to share other data forms from the paper publicly (via figshare or similar service). “available upon request” is problematic for reproducibility, even from the most well-intentioned authors. For example, I no longer have access to some code that former papers from my lab claim “is available upon request”, meaning the code is no longer available to any reader. Putting code on github, data on figshare, etc. ensures the data will be there for future readers.

Response: According to your suggestion, we have deposited the short read data available in GenBank (SRA accession: SRP150515). We also have deposited other sequence data generated in this study in public database and revised the data availability statement in our manuscript.

R3Q4: I don’t like the term microdeletion. 100kb is pretty darned big! Why not just say “deletion”? It is even called a “large genomic change” later (line 244).

Response: According to your and Reviewer 2’s suggestion, we have revised “microdeletion” to “deletion” throughout the text.

5 Line 62: what is the citation for identical mutations in prog1?

Response: We had added the citation. Thank you.

R3Q6: line 154: I don’t know what “no significant differentiation occurred in the sweeping selection between” means. I suspect this is just awkward phrasing?

Response: We have modified the sentence from “This implied that no significant differentiation occurred in the sweeping selection between the two subspecies” to “implying that this region had undergone directional selection during Asian cultivated

rice domestication”. Thank you.

R3Q7: I think the PCR results reported on line 182 conflicts with the phylogeny for site 7 reported on 162. The former suggest all wild rice are similar, but the latter find 8 *rufipogon* that look like cultivated rice. Did I misunderstand something?

Response: At line 182, we described the result from a specific design PCR assay to test whether wild rice samples had the deletion event at the *RPAD* locus. The patterns of PCR bands indicated that all 39 *O. rufipogon* and *O. nivara* accessions carries a large chromosomal segment at the *RPAD* locus similar to that in DXCWR. However, we didn’t sequence these PCR bands to reach polymorphisms at SNP level. At line 162, we describe the sanger sequence results from Site 7 (2,844,613–2,845,874 at the Nipponbare reference genome) from both wild and cultivated rice, which is located at ~1.3-kb downstream of deletion site (2,843,274–2,843,275), and found that eight *O. rufipogon* accessions had a similar sequence with cultivated rice at Site 7, suggesting that the eight accessions might be closer to the direct ancestors in which the deletion event occurred during domestication.

R3Q8: line 352: should say “RPAD”.

Response: We appreciate you finding this error which we have corrected.

R3Q9: line205: I don’t believe you can really call this a “hotspot” without a quantitative comparison to rates of insertion/deletion over the rest of the genome?

Response: We agree with you that we need a quantitative comparison to rates of insertion/deletion over the rest of the genome if we define the *RPAD* locus as is a hot spot of structural variation. Therefore, we had removed the sentence in the revised manuscript.

R3Q10.: line 175: is this really the right citation for the “multiple origins, single

domestication” hypothesis? Isn’t that basically one of the models Tao Sang and Song Ge came up with in 2007?

Response: Thank you for underlining this deficiency. We had revised and added the citations. Thank you.

Reviewers' comments:

Reviewer #1 (Remarks to the Author):

According to the reviewers' comments, the authors have performed further experiments and have made careful modifications on the manuscript. I have no more comment on the revised manuscript and support the acceptance and publication of this manuscript.

Reviewer #2 (Remarks to the Author):

This is my second review of this paper by Wu et al. I find that the authors have addressed most of my prior concerns. I am particularly pleased to see that the authors have been more forthright about information of the germplasm used. The one concern that I feel still could be better addressed is including information in the manuscript about the relationship between PROG1 and the deletion. I agree with the authors that this paper cannot address the causal mutations in PROG1, but given the prominent role proposed for PROG1 in rice domestication, the authors can at least explain if the deletion is in linkage disequilibrium with the proposed causal PROG1 mutations for the domestication phenotype. The information provided about the SNPs in the PROG1 promoter region in the reply to reviewers also seems relevant to readers. It seems that an important next step will be trying to understand whether both PROG1 and the deletion have truly played roles during rice domestication, given that the discovered sweep spans both, and the authors should at least point this out within the manuscript.

Reviewer #3 (Remarks to the Author):

The authors generally do a nice job of addressing my concerns. I have a few remaining comments:

In their response, the authors note "Asian wild rice is heavily admixed with domesticated rice genomes, as indicated that 26% of wild rice samples carrying domesticated prog1 alleles (Genome Research, 27:1029–1038)." If this is indeed the case, how can the authors claim on line 177 that their data support that some rufipogon are closer to the direct wild ancestor?

I was glad to see the authors take advantage of additional sequencing data. I have a few questions regarding this analysis, however. While absence of sequence is not easily interpreted, presence of sequence that either spans the deletion breakpoint or maps to the internal deleted sequence should allow genotyping these lines. The authors claim 57/67 with >1.5X depth supported the deletion. Does this mean sequence from the other 10 definitively showed they did not have the deletion? A bit more careful explanation of these results I think is warranted. I also don't understand the authors request to exclude these from the manuscript. I find these analyses are useful and it seems there is no reason not to include them in the supplement -- they could be mentioned only briefly in the main text in a single line if desired.

The authors are correct that other workers have detected decreased selection in this region. I think those references should be mentioned and cited in the section starting on line 158. As written, it appears the authors are claiming to identify a sweep from their own data alone. While their data are certainly consistent with this model and I believe their result; their data alone are insufficient to make such a claim with high confidence so other work should be cited.

Please review carefully for grammar/phrasing. A few odd phrases remain, such as "sweeping selection region".

Dear Reviewers,

Thank you very much for your helpful comments on our manuscript "Chromosomal deletions within the RPAD locus drove parallel domestication of plant architecture in Asian and African rice" (NCOMMS-17-34535A). We have revised the manuscript accordingly and added two Supplementary Figures (Supplementary Figures 6 and 13) in the revised version. We used red color to mark the modified words and sentences in main text. Our responses to your comments were shown below. We would be grateful if you can kindly be satisfied with the revised manuscript. Thank you again for your time and efforts in taking care of our manuscript.

Sincerely yours,

Lubin Tan, Ph.D

Department of Plant Genetics and Breeding

China Agricultural University, Beijing 100193, China

Responses to reviewer 1's comments:

According to the reviewers' comments, the authors have performed further experiments and have made careful modifications on the manuscript. I have no more comment on the revised manuscript and support the acceptance and publication of this manuscript.

Response: Thank you.

Responses to reviewer 2's comments:

This is my second review of this paper by Wu et al. I find that the authors have addressed most of my prior concerns. I am particularly pleased to see that the authors have been more forthright about information of the germplasm used.

Response: Thank you very much for your positive comments.

R2Q1: The one concern that I feel still could be better addressed is including

information in the manuscript about the relationship between *PROG1* and the deletion. I agree with the authors that this paper cannot address the causal mutations in *PROG1*, but given the prominent role proposed for *PROG1* in rice domestication, the authors can at least explain if the deletion is in linkage disequilibrium with the proposed causal *PROG1* mutations for the domestication phenotype. The information provided about the SNPs in the *PROG1* promoter region in the reply to reviewers also seems relevant to readers. It seems that an important next step will be trying to understand whether both *PROG1* and the deletion have truly played roles during rice domestication, given that the discovered sweep spans both, and the authors should at least point this out within the manuscript.

Response: According to your suggestion, we have added a supplementary file (Supplementary Figure 13) for describing the sequence difference in the *PROG1* promoter region between two parents, DXCWR and GC2. In addition, we have added one paragraph at the Discussion about these information in the revised manuscript.

Responses to reviewer 3's comments:

The authors generally do a nice job of addressing my concerns. I have a few remaining comments:

R3Q1: In their response, the authors note "Asian wild rice is heavily admixed with domesticated rice genomes, as indicated that 26% of wild rice samples carrying domesticated *prog1* alleles (Genome Research, 27:1029–1038)." If this is indeed the case, how can the authors claim on line 177 that their data support that some rufipogon are closer to the direct wild ancestor?

Response: Because the introgressive hybridization between wild and cultivated rice occurred since rice domestication, much living wild rice in Asia is heavily admixed with cultivated rice. Therefore, this is critical to identify the primitive wild rice to study the origin of rice. In this study, we collected 62 accessions of Asian AA-genome wild rice and planted these wild rice accessions in our greenhouse to evaluate the key

domestication traits, including seed shattering, plant architecture, pericarp color, and panicle type. We found that 23 of 62 (37%) accessions of *O. rufipogon* had or was segregating one or several phenotypes like cultivated rice, suggesting that these accessions were admixed with cultivated rice. Therefore, we excluded these 23 accessions in the further analyses and only focused the remaining 39 accessions which had typical wild rice characteristics, especially naturally high levels of shattering and prostrate/semi-prostrate growth habit. Genotyping for the deletion using PCR indicated that all 39 accessions of *O. rufipogon* had a large chromosomal segment at the *RPAD* locus. In addition, the *RPAD* syntenic region in African wild rice *O. barthii* and green foxtail *S. viridis* genomes had a similar tandem repeat of zinc-finger protein-coding genes to that in *O. rufipogon*, suggesting that this is an ancient gene cluster in the Poaceae family. Altogether, we hypothesized that the 39 accessions might belong to the primitive wild rice. Integrating analyses of phenotype evaluation, deletion detection, and phylogenetic analysis, we speculated that the eight of 39 wild rice accessions might be closer to the direct ancestors.

R3Q2: I was glad to see the authors take advantage of additional sequencing data. I have a few questions regarding this analysis, however. While absence of sequence is not easily interpreted, presence of sequence that either spans the deletion breakpoint or maps to the internal deleted sequence should allow genotyping these lines. The authors claim 57/67 with >1.5X depth supported the deletion. Does this mean sequence from the other 10 definitively showed they did not have the deletion? A bit more careful explanation of these results I think is warranted. I also don't understand the authors request to exclude these from the manuscript. I find these analyses are useful and it seems there is no reason not to include them in the supplement -- they could be mentioned only briefly in the main text in a single line if desired.

Response: We didn't obtained the reads covering and spanning the deletion site from those 10 accessions (with >1.5X read depth) in our previous study. According to your

suggestion, we reanalysed the genotypes of the 10 accessions at the *RPAD* locus. The result showed that all 10 accessions have no reads covering and spanning the deletion breakpoints. But there are only a few reads mapping to the internal deleted sequence (repeats masked). Further alignment found that these reads can also map to *japonica* var. Nipponbare reference genome, indicating that these read sequences have other copies in rice genome. All these evidences implied there is deletion in these 10 accessions, but we just don't have directly evidences.

The result of deletion analysis using the public high-through sequencing data have added in the second paragraph of page 6 and Supplementary Fig. 6.

R3Q3: The authors are correct that other workers have detected decreased selection in this region. I think those references should be mentioned and cited in the section starting on line 158. As written, it appears the authors are claiming to identify a sweep from their own data alone. While their data are certainly consistent with this model and I believe their result; their data alone are insufficient to make such a claim with high confidence so other work should be cited.

Response: According to your suggestion, we had revised and added the citation in the revised manuscript.

R3Q4: Please review carefully for grammar/phrasing. A few odd phrases remain, such as "sweeping selection region".

Response: Thanks for pointing this out. We also corrected other grammatical and spelling errors. A few statements were reorganized to make it clear.

Reviewers' comments:

Reviewer #2 (Remarks to the Author):

I thank the authors for addressing my concern about lack of information about the PROG1. While I understand that their QTL interval excluded PROG1 as a causal locus, it may be that the original mapping of PROG1 did not exclude the SPROG1 region as a causal locus. Hence my request that the authors address whether the deletion found by them is in LD with the PROG1 alleles previously proposed as responsible for the difference in plant architecture in cultivated and wild rice. If these polymorphisms are in LD (and I am assuming they are, given that the selective sweep spans both loci), then there is a chance that PROG1 polymorphism does not have a true effect on plant architecture. I don't ask the authors to prove this, but think their last sentence in the new discussion paragraph can be a bit more forthright about the problems in distinguishing whether PROG1 or the deletion at the RPAD locus were the targets of selection during domestication, and the need to better understand the roles of these two loci in affecting plant architecture and during domestication.

Please look over the new text for small typos. In the new figure and legend, for example, marker is misspelled twice.

Reviewer #3 (Remarks to the Author):

The authors have addressed most of my concerns. I still have trouble understanding how we can have confidence in which wild samples are ancestral if there is thought to be rampant gene flow. Yes, these samples don't have wild alleles at a few loci tested, but they may have domesticated rice alleles in many other regions of the genome. I'd want to see a more formal analysis that takes into account gene flow (treemix, admixture graphs, IBD analysis, something) when determining ancestry. I would prefer to see this claim dropped, or at the very least add the caveat that the relationship seen on the tree could be due to introgression.

Dear Reviewers,

Thank you very much again for your helpful comments on our manuscript (NCOMMS-17-34535). According to your suggestions, we have pointed out in the discussion section that whether PROG1 or the deletion at the RPAD locus were the targets of selection during domestication need to be addressed in future studies, and removed the statement that some *O. rufipogon* are closer to the direct wild ancestor from the manuscript. Our responses to your comments are described below. We would be grateful if you can kindly be satisfied with the revised manuscript.

Sincerely yours,

Lubin Tan, Ph.D

Department of Plant Genetics and Breeding

China Agricultural University, Beijing 100193, China

Responses to reviewer 2's comments:

R2Q1: I thank the authors for addressing my concern about lack of information about the PROG1. While I understand that their QTL interval excluded PROG1 as a causal locus, it may be that the original mapping of PROG1 did not exclude the SPROG1 region as a causal locus. Hence my request that the authors address whether the deletion found by them is in LD with the PROG1 alleles previously proposed as responsible for the difference in plant architecture in cultivated and wild rice. If these polymorphism are in LD (and I am assuming they are, given that the selective sweep spans both loci), then there is a chance that PROG1 polymorphism does not have a true effect on plant architecture. I don't ask the authors to prove this, but think their last sentence in the new discussion paragraph can be a bit more forthright about the problems in distinguishing whether PROG1 or the deletion at the RPAD locus were the targets of selection during domestication, and the need to better understand the roles of these two loci in affecting plant architecture and during domestication.

Response: We agree with the reviewer. It is an important next step to distinguish whether *PROG1* or the deletion at the *RPAD* locus were the targets of selection during domestication and understand the roles of these two loci in rice plant architecture domestication. We now point out in the discussion section that the question need to be addressed in future studies (lines 267-272).

R2Q2: Please look over the new text for small typos. In the new figure and legend, for example, marker is misspelled twice.

Response: Thanks for pointing this out. We have corrected the spelling errors in the figure legend of Supplementary Fig. 13.

Responses to reviewer 3's comments:

The authors have addressed most of my concerns. I still have trouble understanding how we can have confidence in which wild samples are ancestral if there is thought to be rampant gene flow. Yes, these samples don't have wild alleles at a few loci tested, but they may have domesticated rice alleles in many other regions of the genome. I'd want to see a more formal analysis that takes into account gene flow (treemix, admixture graphs, IBD analysis, something) when determining ancestry. I would prefer to see this claim dropped, or at the very least add the caveat that the relationship seen on the tree could be due to introgression.

Response: We agree with the reviewer. However, we do not have genome-wide sequences of all surveyed samples in this study for analysing gene flow and determining ancestry. Hence, according to your suggestion, we have removed the sentence “Eight accessions of *O. rufipogon* from China dispersed into this same clade of cultivated rice, suggesting that these wild accessions might be closer to the direct ancestors in which the deletion event occurred during domestication” (line 182, page 7) in the revised manuscript. Thank you.